# Cattle management in an Iron Age/Roman settlement in the Netherlands: Archaeozoological and stable isotope analysis

**Maaike Groot[1]\*, Umberto Albarella[2], Jana Eger[3], Jane Evans[4]**

**1** Institut für Prähistorische Archäologie, Freie Universität, Berlin, Germany, **2** Department of Archaeology, University of Sheffield, Sheffield, United Kingdom, **3** Eurasien-Abteilung, Deutsches Archäologisches Institut, Berlin, Germany, **4** NERC Isotope Geosciences Laboratory, British Geological Survey, Environmental Science Centre, Keyworth, Nottingham, United Kingdom

\* maaike.groot@fu-berlin.de

**Data Availability Statement:** All relevant data are within the paper and its Supporting information files.

## Abstract

Cattle were the predominant domestic animal in the Iron Age and Roman Netherlands, yet their management is still incompletely understood. Some aspects of cattle management, such as birth season and the provision of fodder, have received little or no attention so far. This paper is the first to investigate these aspects for the Iron Age and Roman Netherlands, through a case study of the site of Houten-Castellum. The rural settlement of Houten-Castellum was inhabited from the Middle Iron Age to the Middle Roman period, allowing a comparison between the Iron Age and Roman period. Excavations at this site have yielded a large, well-preserved animal bone assemblage. This paper investigates cattle husbandry by using an integrated approach, combining a multi-isotope analysis (oxygen, carbon and strontium) with archaeozoological and archaeobotanical results from Houten-Castellum and comparing the results with archaeobotanical evidence for fodder and evidence for dairy use for the Iron Age and Roman Netherlands in general. While our data set is small and results must therefore be interpreted cautiously, there is convincing evidence for an extended birth season in the Middle Iron Age, as well as the use of fodder.

## Introduction

Iron Age society in the central part of the Netherlands was based on mixed farming with a strong pastoral component [1]. This ties in with the river landscape, which offered limited space for growing cereals and other crops but plentiful pasture for livestock [2]. Society was egalitarian and self-sufficient in terms of food [1, 3, 4]. Archaeozoological research has shown the predominance of cattle compared to other farm animals [1, 5, 6]. Cattle supported arable farming by traction and manure, provided food in the form of meat and dairy products and raw materials for clothing and artefacts; they also had an important social and status significance [6, 7]. Recent studies have shown that mobility of livestock, including cattle, played a significant role in the study area during the Iron Age [8, 9]. Mobility can be related to the role of

**Funding:** This work was supported by a Marie Sklodowska-Curie grant funded by the European Union's Horizon 2020 research and innovation programme (https://ec.europa.eu/programmes/horizon2020/en/home) (Grant agreement No 740394) (MG). We acknowledge support by the Open Access Publication Fund of the Freie Universität Berlin. The funders had no role in study design, data collection and analysis, decision to publish, or preparation of the manuscript.

**Competing interests:** The authors have declared that no competing interests exist.

cattle as an exchange medium, cattle raiding and the movement of human groups—almost certainly with their livestock [10].

The incorporation of the southern half of the Netherlands into the Roman Empire had a major and widespread impact on farming; this included the introduction of new crops and animal species, a size increase in livestock, intensification of agriculture, a higher degree of specialisation and better organisation of storage and transport of surplus goods [2, 11, 12]. The agrarian economy was transformed from mainly self-sufficient to surplus-producing. An early monetary economy facilitated transactions between the agricultural producers and the urban and military consumers, and a more distinct separation between town and countryside took place. There is evidence that some food was supplied from outside the region [13–15], but most of the meat is assumed to have come from local supply, with cattle as the main meat provider in the Roman towns and army camps. Cattle were also important as transport animals. There is little evidence for mobility towards the rural site of Houten-Castellum during the Roman period, which is not unexpected, as livestock is more likely to have left the settlement to supply army camps or towns [8]. This is supported by strontium isotope results for cattle from the Roman town of Heerlen, which show a high degree of mobility for cattle, which would have supplied the town with meat, but probably also included draught cattle sent for slaughter at the end of their working lives [8].

Although cattle were hugely important in both Iron Age and Roman societies, their day-to-day management is still incompletely understood. The hypsodont structure (high-crowned with enamel extending beyond the gum line) of the herbivore tooth provides the opportunity for high-resolution analysis of seasonal behaviour, which is commonly used in dietary and seasonality studies [16, 17]. Stable isotope analyses in the Netherlands have so far mostly focused on humans [18–21], with a few exceptions [18, 22]. This paper will investigate cattle management in the Iron Age and Roman Netherlands through stable isotope analysis (oxygen and carbon) of cattle teeth from the rural settlement of Houten-Castellum. The results will be integrated with previously published strontium isotope analysis carried out for the same site [8], resulting in a multi-isotope analysis. This will provide insight into aspects of cattle management that are hard to investigate through other methods. Our main questions for this paper are:

- Was there a limited or extended birth season for cattle? What evidence is there for dairying in the site of Houten-Castellum?

- What information do carbon isotopes from cattle tooth enamel provide on seasonal variation in cattle diet? Do they indicate foddering of cattle? Do they indicate the type of pasture?

The results from the stable isotope analysis will be combined with archaeozoological results from Houten-Castellum, which provide further information on cattle exploitation. We will then look at archaeobotanical evidence for cattle diet and fodder and evidence for dairy use (cattle mortality profiles and sex ratios and organic residue analysis), both at Houten-Castellum and more generally, in the Iron Age and Roman Netherlands.

## Material and methods

### Archaeozoological methods

To establish mortality profiles for cattle from Houten-Castellum, both mandibular tooth eruption and wear and epiphyseal fusion data were considered. Grant's [23] method was used for tooth eruption and wear. Only complete mandibles were included. The mandible wear stages were assigned to the age categories defined by Halstead [24, 25]. Epiphyseal fusion data were

assigned to different stages to calculate the percentage of animals killed in five categories: younger than 1 year, 1–2 years, 2–3 years, 3–4 years and older than 4 years [26–29]. Sex ratios were established in two ways. First, the shape of the pubic bone was used to assign fragments to males or females and second, biometrically through plotting the distal breadth of the trochlea of the humerus and assessing the distribution. The results from the Early and Middle Roman period have been combined since the Middle Roman phase only covers the early part of that period (until circa AD 120).

## Theoretical principles behind oxygen and carbon

The $\delta^{18}$oxygen values in the carbonate fraction of tooth enamel are determined by the values in drinking water and water in food, which are related to values in precipitation [30]. The isotopic composition of precipitation is determined by several factors, including temperature, distance from the coast, latitude, altitude and the amount of precipitation [31, 32]. Since these factors vary geographically, $\delta^{18}$O values also vary geographically. In addition to geographical variation, $\delta^{18}$O values vary seasonally. In a temperate climate, summers are typically warmer and drier (leading to higher $\delta^{18}$O values) and winters colder and wetter (leading to lower $\delta^{18}$O values). The $\delta^{18}$O values in the carbonate fraction of tooth enamel reflect climatic conditions during tooth formation. The hypsodont cheek teeth of herbivores allow sequential sampling of enamel from top (occlusal surface—where tooth formation starts) to bottom (the enamel-root junction; ERJ—where tooth formation ends), which provides a record of $\delta^{18}$O values over the period during which the tooth formed. This can provide insight into seasonal changes, not just of the $\delta^{18}$O values, but also of conjointly measured $\delta^{13}$carbon values. Sequential sampling of molars is generally expected to show a sinusoidal curve of $\delta^{18}$O values, with the maximum reflecting summer and the minimum winter.

The $\delta^{13}$carbon values in the carbonate fraction of tooth enamel reflect diet, which for cattle consists of vegetation. The $\delta^{13}$C values of plants depends on the type of photosynthetic pathway ($C_3$ or $C_4$ plants), the part of the plant, water availability/aridity, salinity, altitude and season [33, 34]. In addition, the canopy effect leads to lower $\delta^{13}$C values in plants growing in closed forest [33, 35]. The mean $\delta^{13}$C value for $C_4$ plants is around -13‰, while it is around -27‰ for $C_3$ plants [34–37]. The $\delta^{13}$C enrichment from diet to tooth enamel for large ruminants in general and cattle specifically has been estimated at 14.1‰ to 16‰ [38–40]. Sequential sampling for $\delta^{13}$C values in cattle teeth will show seasonal changes in vegetation in free-ranging cattle, with the $\delta^{13}$C curve following the $\delta^{18}$O in temperate climates. Providing cattle with fodder or moving them to different pastures during an annual cycle can change the curve [41–45]. Flat curves suggest a uniform food supply during tooth formation [46] and could suggest the provision of fodder.

## Balasse's method

The $\delta^{18}$O values from sequential sampled tooth enamel can also be used to infer the length of the birthing period or to express birth seasonality for archaeological animals [47–50]. Since the timing of tooth growth is fixed within a species, inter-individual variability in the $\delta^{18}$O curve permits to investigate the distribution of birth in past herds. In order to estimate the birth season of cattle in this study ($x_0/X$), the position (distance from ERJ) of the maximum $\delta^{18}$O value ($x_0$) were normalized to the periodic cycle corresponding to the crown length formed potentially over a year (X). This technique eliminates differences in tooth size between individual animals. To allow an objective description, the sinusoidal $\delta^{18}$O sequences of the sample were modelled using the method developed by Balasse and colleagues [47, 48]. The Pearson correlation coefficient was used to measure the proximity between the modelled and raw $\delta^{18}$O data.

### The chemical preparation and isotope analysis of carbon and oxygen in structural carbonate

For the isotope analysis of phosphate carbonate oxygen, approximately 3 milligrams of prepared enamel was loaded into a glass vial and sealed with septa. The vials are transferred to a hot block at 90°C on the GV Multiprep system. The vials are evacuated and 4 drops of anhydrous phosphoric acid are added. The resultant $CO_2$ was collected cryogenically for 14 minutes and transferred to a GV IsoPrime dual inlet mass spectrometer. The resultant isotope values are treated as a carbonate. $\delta^{18}O$ is reported as per mil (‰) ($^{18}O/^{16}O$) normalized to the PDB scale using a within-run calcite laboratory standard (KCM) calibrated against SRM19, NIST reference material and were converted to the SMOW scale using the published conversion equation of Coplen [51]: SMOW = (1.03091 x $\delta^{18}O_{VPDB}$) +30.91. Analytical reproducibility for this run of laboratory standard calcite (KCM) is for $\delta^{18}O_{SMOW}$ = ± 0.06‰ (1 σ, n = 40) and $\delta^{13}_{VPDB}$ is ± 0.03‰ (1s, n = 40). The reproducibility of the enamel analysis, based on reproducibility of duplicate pairs of samples, for $\delta^{18}O_{VSMOW}$ is ± 0.04‰ (1SD, n = 4).

### Samples and sample preparation

A total of 10 cattle teeth are included in this study, all from the site of Houten-Castellum (Fig 1). Four left and six right lower third molars from 10 different individuals were selected. S1 Table shows the context information of the selected teeth. Teeth were selected based on the length of the crown and a roughly equal distribution over the different time periods. The 10 teeth are dated as follows: 3 from the Middle Iron Age (500–250 BC), 3 from the Late Iron Age (250–19 BC) and 4 from the Early/Middle Roman period (19 BC-AD 120). The contexts from which the teeth derive were dated based on stratigraphy, typochronology for pottery and metal finds and $^{14}C$ dating on wood [52]. Two molars were excluded from modelling of $\delta^{18}O$ sequences due to the lack of reliable maximum and minimum $\delta^{18}$oxygen values (MG13 and MG24).

After cleaning the surface of the teeth with a dental drill, a strip of enamel was taken from the buccal side of the anterior lobe. Sequential sampling for stable isotope analysis of oxygen and carbon was carried out by taking transversal slices along the length of the strip, from apex to ERJ. These chips of enamel were then ground into powder.

Cattle lower third molars are formed between 9 and 23 months [53], with another 6 months before mineralisation is complete [16]. This means that the samples provide information on the second and early third year of life, with the oldest samples present at the apex of the tooth, which is formed first.

## Archaeological background and earlier stable isotope research

### Background Houten-Castellum

The archaeological site of Houten-Castellum was a rural settlement inhabited in the Iron Age and Roman period [54]. It is located in the central Dutch river area, within the border of the Roman Empire (Fig 1). The site consists of a residual channel containing settlement refuse and intentional deposits, as well as traces of habitation on the western bank of the channel. It seems to have been abandoned by the end of the 2nd century AD. The large quantity of finds includes ca. 86,000 animal bone remains [55]. Most of the finds come from layers within the residual channel, which can often be dated accurately. Due to the waterlogged conditions, the preservation of the animal bones is excellent. The site of Houten-Castellum was a self-sufficient agrarian community, probably producing a surplus of food in the Roman period. Cattle is the main animal species in all periods, with the highest proportion occurring in the Middle

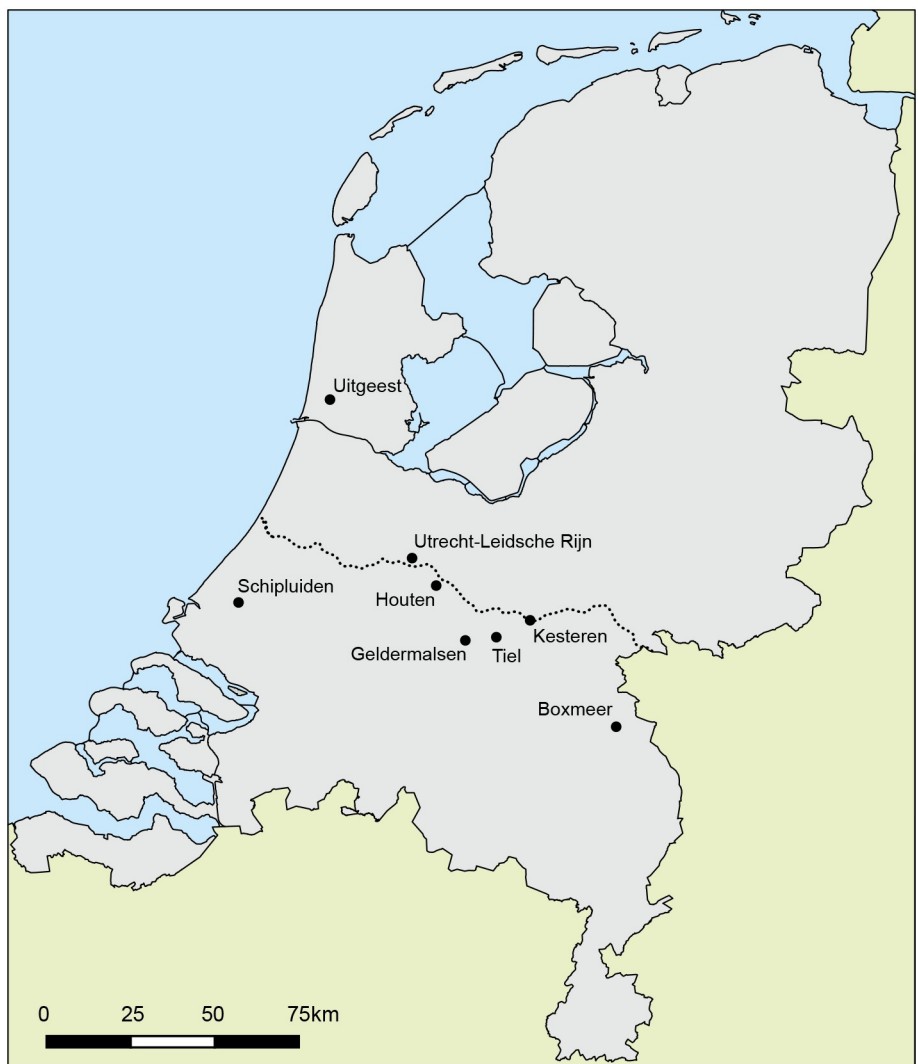

**Fig 1. Map of the Netherlands.** Sites mentioned in the text are marked. The dotted line represents the Roman border (Map by Marjolein Haars, BCL Archaeological Support).

Iron Age [55]. Palynological data suggest that the landscape around Houten was open with scattered groves of trees in the early Middle Iron Age [56]. A decline in tree pollen indicates that the landscape became more open over time and consisted mostly of grassland and arable fields.

### Earlier research: Stable oxygen and carbon isotope analysis from cattle tooth enamel in the Netherlands

The only archaeological site in the Netherlands for which sequential $\delta^{18}O$ and $\delta^{13}C$ analysis of cattle tooth enamel has been published is Neolithic Schipluiden [50]. The data were combined with $\delta^{13}C$ and $\delta^{15}N$ isotope analysis of cattle bone collagen and archaeozoological analysis with the aim to investigate cattle diet and calving season in one of the first permanently settled sites in the Netherlands. Lower third molars from eight cattle were sampled. For seven teeth, the seasonal curve was complete enough to establish the season of birth. They show that cattle

births occurred over a period of 5.5 months. The extended calving season would have resulted in a year-round availability of milk.

The $\delta^{13}$C values for some of the Schipluiden cattle are lower than expected. Since such low values were not found in red deer and suids, they have been interpreted as evidence of the use of leafy fodder rather than grazing on waterlogged pastures [50]. In one animal, the minimum $\delta^{13}$C value coincided with the minimum $\delta^{18}$O value, showing that this value was reached in winter. Overall, the authors of the Schipluiden study believe that some cattle occasionally received leafy fodder in winter to support lactating cows and thereby extending the lactation period.

## Archaeological and archaeobotanical evidence for cattle pasture and fodder

To understand cattle diet, we must investigate the potential for livestock pasture in the river area in general and more specifically in the surroundings of Houten-Castellum. Archaeobotanical research for Houten-Castellum and other settlements provide indications for landscape, vegetation and land use, as well as indications for livestock diet and the use of fodder. The use of fodder can also be deduced indirectly from the presence and size of stables and storage structures. The characteristic farmhouse in the Iron Age and Roman Netherlands is the byre-house, which contains a stable section. In regions where wood has been preserved, stable boxes suggest that it was mainly larger livestock that was stabled. Stabled cattle would have required fodder.

Most natural woodland in the river area had disappeared before the Middle Bronze Age, resulting in an open landscape with only small patches of woodland [57, 58]. Streamridges (former river systems consisting of a river bed and river banks) were the highest parts of the landscape, with the least risk of flooding. They were used for habitation and arable fields. They are unlikely to have offered space for pasture, but cattle could have grazed stubble fields after harvest and fallow fields (weeds and crops grown from spilled seed from previous years). Botanical research for a site close to Houten-Castellum (Houten-Tiellandt) found indications for a rotation system with fallow years [59]. Insect remains from the same site suggest that livestock grazed on stubble fields [59]. Flooding reduced the available pasture in the low-lying flood basins in winter, but in summer, they provided fertile pasture for livestock. The flood basins consisted of reed marshes and damp grassland [2]. The grassland could also be used to collect hay. Pollen from mineralised manure from either cattle or pig from the settlement of Kesteren-De Woerd shows that the manure's producer grazed on damp grassland or reed marshes, almost certainly in the flood basins [59, 60]. Remains of water plants and grassland plants found in wells in Geldermalsen-Hondsgemet could stem from either fodder or manure [61]. If fodder, then it is most likely that this was fed in summer, since water plants are not suitable for long-term storage [2]. If manure, then the plant remains indicate the type of pasture: marshy areas in the flood basins and dry grassland on streamridges or levees [61]. An additional indication of livestock diet is provided by the high proportions of water and marsh plants (compared to weeds, meadow plants and woodland plants) found in the Roman settlements of Tiel-Passewaaijse Hogeweg, Geldermalsen-Hondsgemet and Kesteren-De Woerd [2], which again can be explained by manure or fodder. Evidence for fodder was found in Wijk bij Duurstede-De Horden, where it consisted of a combination of hay, uncleaned cereals and weeds [62].

The large storage capacity in the settlement of Tiel-Passewaaijse Hogeweg (much larger than the local arable fields could supply) has been taken as an indication for the possibility of the storage of fodder [2]. The systematic use of branches and leaves as fodder is unlikely since woodland was scarce. Livestock is also unlikely to have been fed with cereals. The area available

for growing cereals and other crops was limited, and any surplus is more likely to have been sold [2].

Archaeobotanical research (pollen and macro-remains) in Houten-Castellum indicates that the landscape was partly open with patches of woodland at the start of the Middle Iron Age and became progressively less wooded and more intensively used during the Middle and Late Iron Age [56]. Grasslands were an important landscape element, increasing over time. There are also indications for arable fields, which—in the Roman period at least—were on fertile soil and contained damp parts. The latter means that cereals are likely to have been grown as a summer crop, leaving stubble fields as a possible livestock food source in winter.

## Indications for milk: Organic residue analysis

The use of milk can be attested through organic residue analysis of pottery, distinguishing between lipids from milk fat and body fat. Unfortunately, such research has not yet been carried out systematically for the Iron Age and Roman Netherlands. One exception, where animal protein could stem from milk, is represented by the Early Roman site of Utrecht-Leidsche Rijn 60, where organic residues from cooking pots and bowls were analysed [63]. A combination of animal protein, animal or vegetable fats and green plant parts was interpreted as either cheese or fermented milk products with herbs, or fermented fish products with leaf vegetables. Organic residue analyses for other sites in the region (Late Iron Age/Roman Tiel-Medel-Afronding and Roman Boxmeer-Sterckwijk) have not found evidence for dairy use [64, 65]. Organic residue analysis of ceramics from the Roman-period site of Uitgeest-Groot Dorregeest has identified charred animal products that are likely to be milk [66]. In summary, organic residue analysis shows evidence for dairy, but it does not seem to have been ubiquitous. However, research so far has been limited, and further research may change this impression.

## Archaeozoological evidence for dairy use in the Iron Age and Roman Netherlands

Cattle mortality profiles for the Iron Age and Roman Netherlands show that few sites focused on the exploitation of milk (Fig 2). Dairy exploitation is found most often in the western part of the Netherlands. Dairy exploitation decreases in the Roman period, which is not just suggested by mortality profiles, but also by an increase in the proportion of adult male cattle. Nevertheless, even when exploitation is aimed more at other cattle products, this does not exclude the use of milk; it only means that this was not the primary exploitation aim.

## Results

### Archaeozoology: Mortality profiles and sex ratios of cattle

Mortality profiles for cattle at Houten-Castellum, based on mandibular tooth eruption and wear, show that the largest slaughter peak occurred between 18 and 30 months (Fig 3). Mortality of young calves (0–1 months) was low: 5–8%. A higher proportion (12–17%) was killed between 8 and 18 months. There is an increase in adult and older cattle over time.

Mortality profiles based on epiphyseal fusion are only available for the Middle Iron Age and Roman period. They show mortality in the first two years of life of 10% or less, higher slaughter rates between 2 and 4 years and high survival into adulthood (Fig 4).

Sex ratios based on the shape of the pubic bone show both a predominance of females and a higher female to male ratio in the Middle Iron Age than in the Roman period (Fig 5; unfortunately, there are no data for the Late Iron Age). Measurements of the distal humerus also show

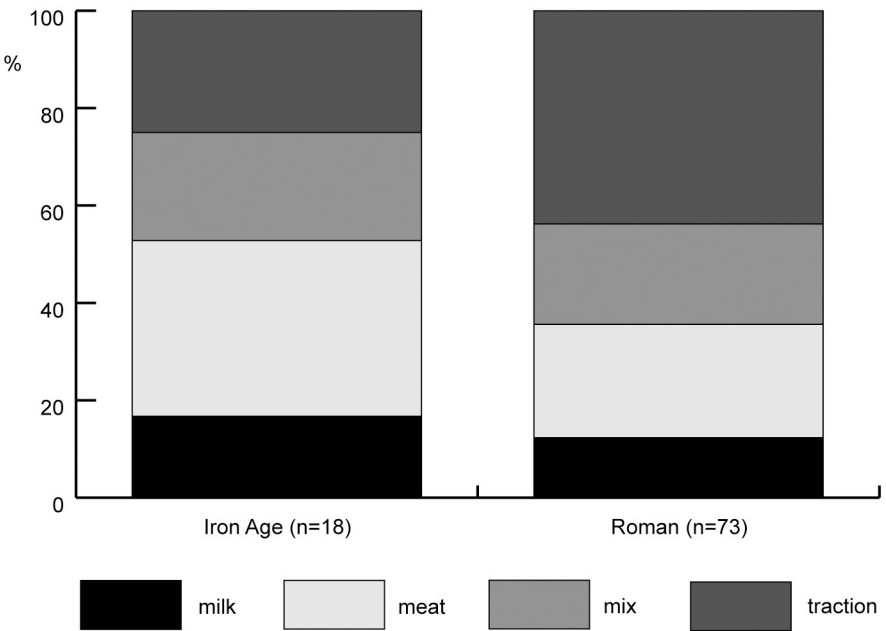

**Fig 2. Percentages of assemblages per type of cattle exploitation for the Iron Age and Roman Netherlands (all regions and types of site combined).** n = number of sites for which the type of exploitation could be established. Mortality profiles based on tooth eruption and wear and epiphyseal fusion were analysed to establish the type of exploitation.

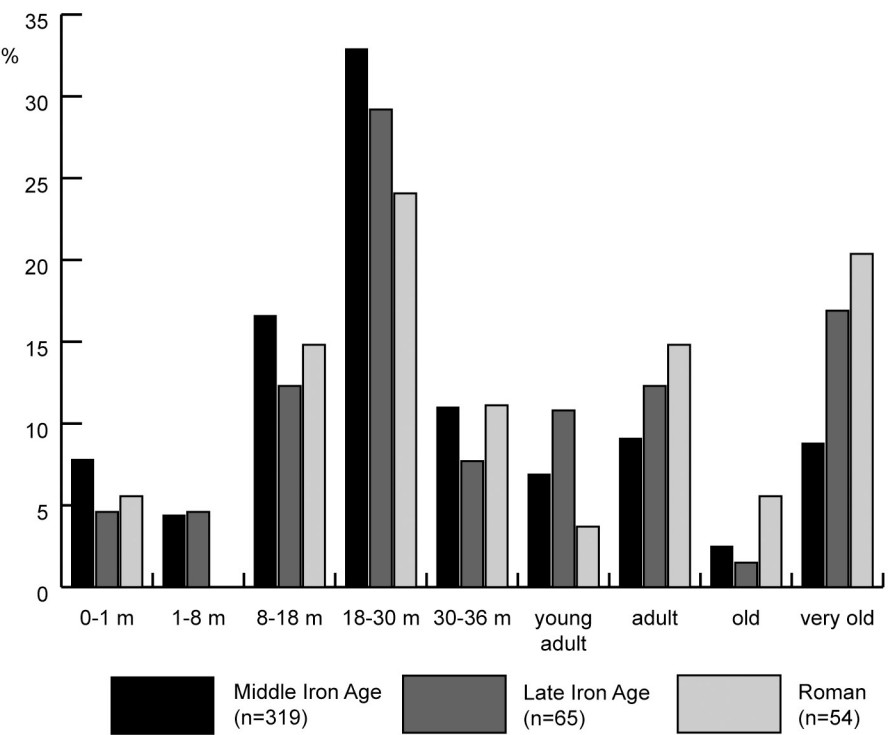

**Fig 3. Mortality profile for cattle from Houten-Castellum, based on mandibular tooth eruption and wear.**

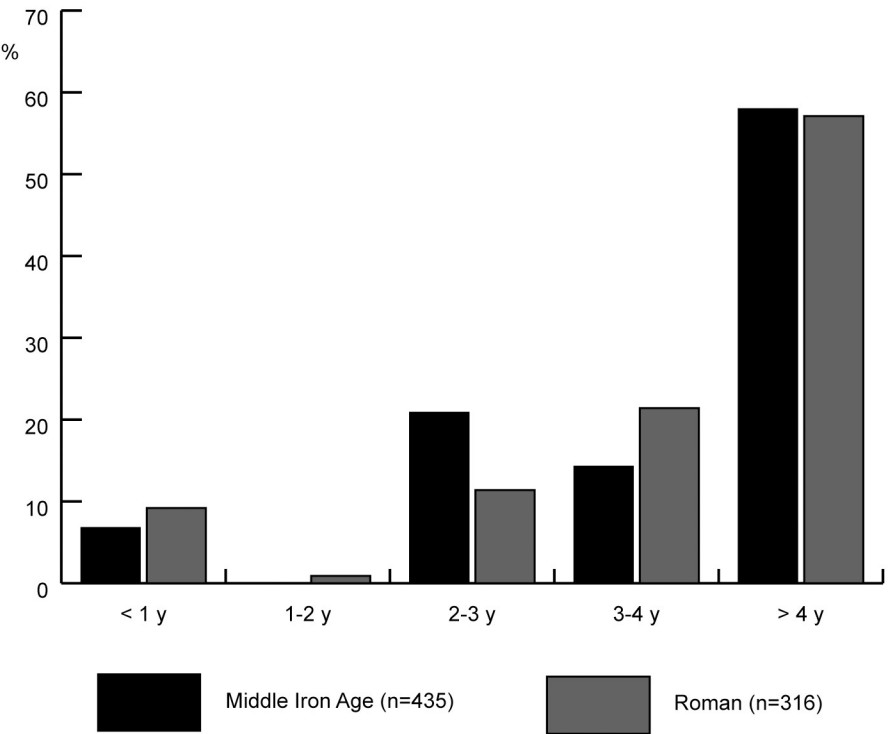

**Fig 4. Mortality profile for cattle from Houten-Castellum, based on epiphyseal fusion.**

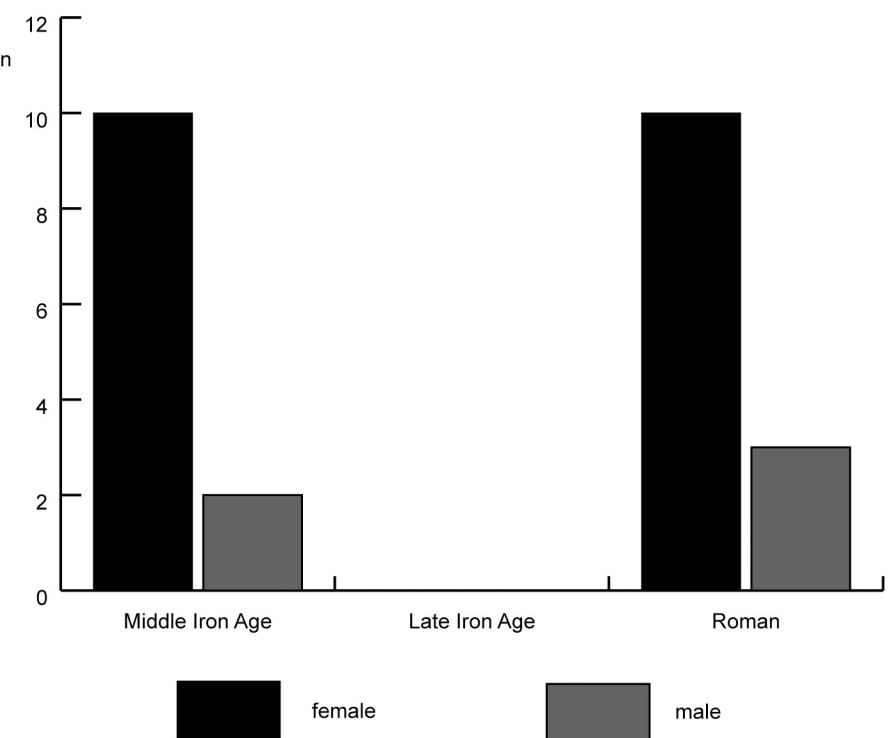

**Fig 5. Sex ratios for cattle from Houten-Castellum based on the shape of the pubic bone.**

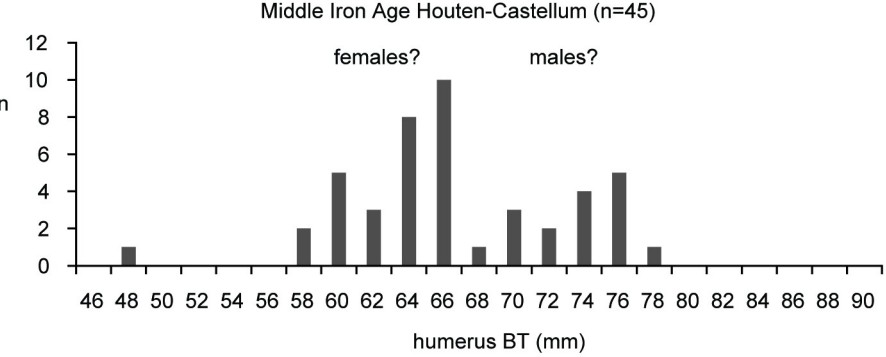

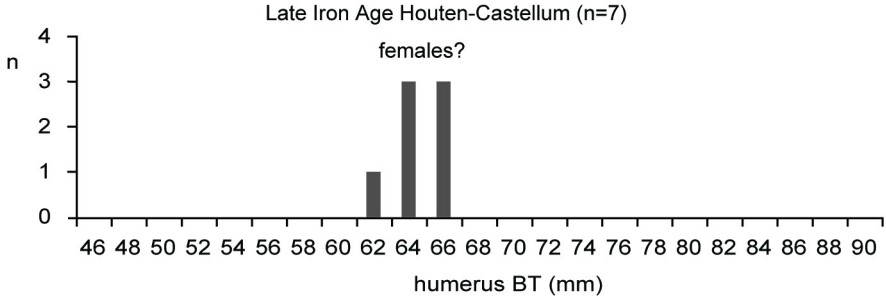

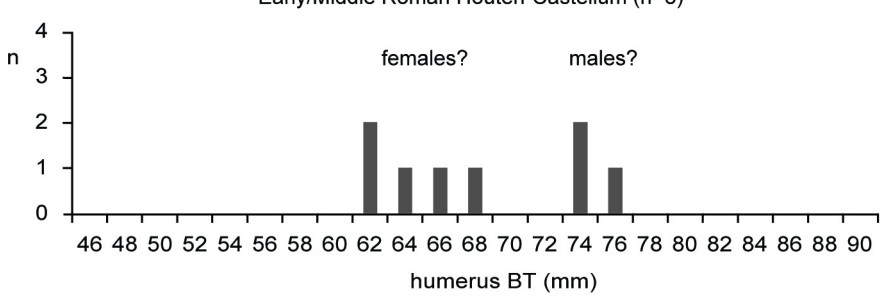

**Fig 6. Houten-Castellum.** Cattle: breadth of the trochlea of the humerus.

a predominance of females in all periods, and although the sample size is small, the proportion of males seems to increase in the Roman period (Fig 6).

## Stable isotope analysis

Of the 10 teeth, two were identified as coming from non-local cattle by strontium isotope analysis: MG31 (Middle Iron Age) and MG04 (Late Iron Age) [8]. The other eight samples were consistent with the local signal. The results of the stable isotope analysis are summarised in Table 1 and plotted in Fig 7. All $\delta^{18}O$ and $\delta^{13}C$ values per tooth are reported in S2 Table. Non-local MG04 has $\delta^{18}O$ and $\delta^{13}C$ values that deviate from the other teeth, while MG31 does not. The $\delta^{18}O$ values vary between 22.72 and 28.23‰ with an intra-tooth amplitude of variation of 1.62 to 2.91‰ (Table 1). The $\delta^{13}C$ values vary between -13.86 and -11.36‰, with an intra-

**Table 1. Summary statistics for oxygen (δ18O) and carbon (δ13C) isotope values for cattle from Houten-Castellum.**

| | δ18O$_{VSMOW}$ | | | | δ13C$_{VPDB}$ | | | |
|---|---|---|---|---|---|---|---|---|
| | minimum | maximum | mean | amplitude | minimum | maximum | mean | amplitude |
| MG04 (non-local) | 25.32 | 28.23 | 26.54 | 2.91 | -13.86 | -12.94 | -13.38 | 0.92 |
| MG06 | 23.77 | 25.39 | 24.67 | 1.62 | -12.61 | -12.10 | -12.27 | 0.51 |
| MG08 | 24.24 | 25.96 | 24.99 | 1.72 | -12.11 | -11.57 | -11.90 | 0.54 |
| MG13 | 23.98 | 25.96 | 24.54 | 1.98 | -13.76 | -11.49 | -11.94 | 2.27 |
| MG24 | 22.82 | 24.56 | 23.39 | 1.74 | -12.09 | -11.88 | -12.00 | 0.21 |
| MG26 | 24.24 | 26.99 | 25.53 | 2.75 | -12.22 | -11.65 | -11.99 | 0.57 |
| MG31 (non-local) | 24.09 | 26.52 | 25.30 | 2.43 | -12.84 | -11.62 | -12.22 | 1.22 |
| MG33 | 22.72 | 24.72 | 23.69 | 2.00 | -11.9 | -11.36 | -11.65 | 0.54 |
| MG38 | 23.47 | 25.25 | 24.43 | 1.78 | -12.59 | -11.79 | -12.23 | 0.80 |
| MG40 | 23.71 | 26.08 | 24.68 | 2.37 | -12.76 | -12.37 | -12.62 | 0.40 |

Amplitude: intra-tooth peak-to-peak amplitude of variation.

tooth amplitude of variation between 0.21 and 0.92‰ for eight animals and two higher values of 1.22‰ and 2.27‰ (both non-local animals) (Table 1).

## Modelling of δ$^{18}$O sequences measured in cattle third molars

Results for the best fit for combined variation of the equation parameters are shown in Table 2. The location of the maximum δ$^{18}$O value in tooth crown ($x_o$) varies from 0.5 to 22.5 mm from ERJ. Pearson's r-values (between 0.87 and 0.98) confirm the similarity between the δ$^{18}$O values and the modelled data. Middle Iron Age values are distributed between 0.03 and 0.74 (range of 0.71) in three specimens, born within 71% of a yearly cycle, corresponding to a birthing season of approximately eight and a half months (Fig 8). For MG06, the location of the maximum δ$^{18}$O value in tooth crown ($x_o$) is exceptional compared to all other samples (indicating a different birth season); however, although Pearson's r-value is lower than that for the other samples, it is still high enough for this tooth to be included. Late Iron Age values are distributed between 0.01 and 0.28 (range of 0.27) in two specimens, born within 27% of a yearly cycle, corresponding to a birthing season of approximately three months. Roman values are distributed between 0.09 and 0.24 (range of 0.15) in three specimens, born within 15% of a yearly cycle, corresponding to a birthing season of approximately two months.

## Discussion

### Birth season

Free-ranging cattle in northwestern Europe would naturally have a limited birth season of 2–3 months, from May to June/July [47]. Our results show a birth season of 8.5 months in the Middle Iron Age, 3 months in the Late Iron Age and 2 months in the Roman period. The number of teeth sampled per period is very small, so that these figures should be seen as minimum figures. An extended season of birth suggests a deliberate extension of the period during which cows can be milked.

Organic residue analysis shows that milk or milk products were used in the Iron Age and Roman Netherlands, but dairy does not seem to have been ubiquitous. Mortality profiles for cattle in the Iron Age and Roman Netherlands show only a few sites with specialised milk production. Mortality profiles for Houten-Castellum do not show a focus on dairy products (the

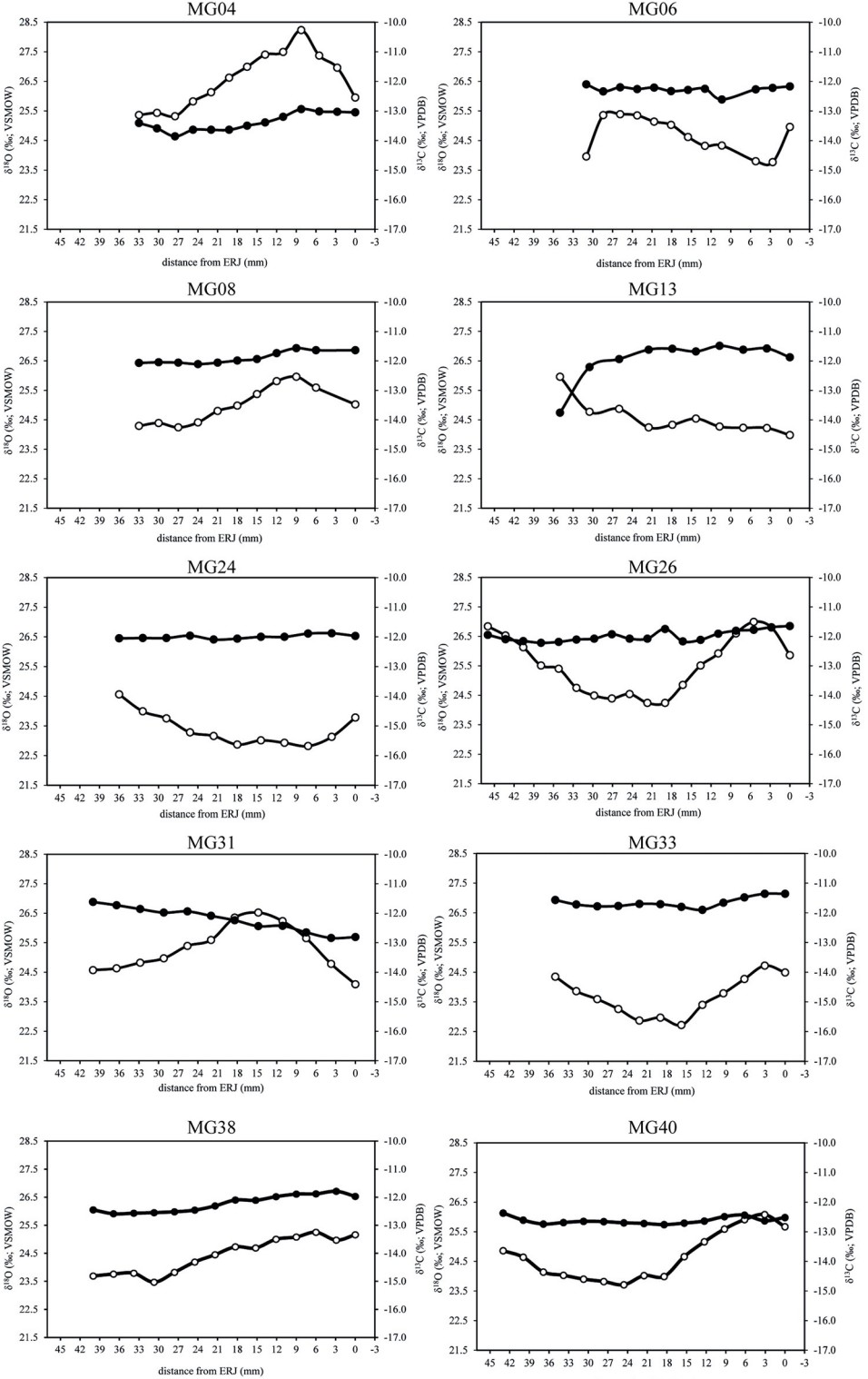

**Fig 7. Results from the sequential analysis of oxygen (δ¹⁸O, white circles) and carbon (δ¹³C, black circles) isotope composition in sampled enamel bioapatite from 10 cattle lower third molars (M3).**

**Table 2. Results from the calculation of the best fit (method of least squares, applying the microsoft excel software solver function) between the modelled and measured δ18O datasets.**

| Specimen | X (mm) | A (‰) | $x_0$ (mm) | M (‰) | $x_0/X$ | r (Pearson) | Arch. Period |
|---|---|---|---|---|---|---|---|
| MG04 | 37.8 | 1.2 | 10.4 | 26.5 | 0.28 | 0.97 | LIA |
| MG06 | 30.3 | 0.8 | 22.5 | 24.6 | 0.74 | 0.87 | MIA |
| MG08 | 38.4 | 0.8 | 9.3 | 25.0 | 0.24 | 0.98 | Roman |
| MG26 | 41.7 | 1.2 | 4.3 | 25.4 | 0.10 | 0.97 | Roman |
| MG31 | 37.9 | 1.0 | 16.2 | 25.4 | 0.43 | 0.95 | MIA |
| MG33 | 38.7 | 0.9 | 0.5 | 23.7 | 0.01 | 0.98 | LIA |
| MG38 | 59.8 | 0.8 | 5.7 | 24.4 | 0.09 | 0.98 | Roman |
| MG40 | 53.0 | 1.1 | 1.5 | 24.8 | 0.03 | 0.98 | MIA |

X: period of the cycle/length of tooth crown potentially formed over one annual cycle; A: amplitude; M: mean; $x_0$: position in the tooth crown where δ18O has the highest value; r: the proximity between the modelled data and the measured δ18O values.

The modelling uses the equation from Balasse and colleagues [47]: $\delta^{18}O_{model} = A \cdot \cos(2\Pi (x - x_0)/X) + M$.

ones based on epiphyseal fusion even less so). The main slaughter peak between 18–30 months represents slaughter for meat of cattle whose growth had started slowing down. Mortality of calves younger than 1 month old is consistent with natural mortality without any need to resort to dairying as an explanation. The presence of this age category, as well as the presence of bones from foetal and neonatal animals, suggests that calving took place within or around the settlement. Cattle killed between 8 and 18 months (12 to 17%) could represent post-lactation slaughter. The increase in older cattle over time is a phenomenon that has been observed

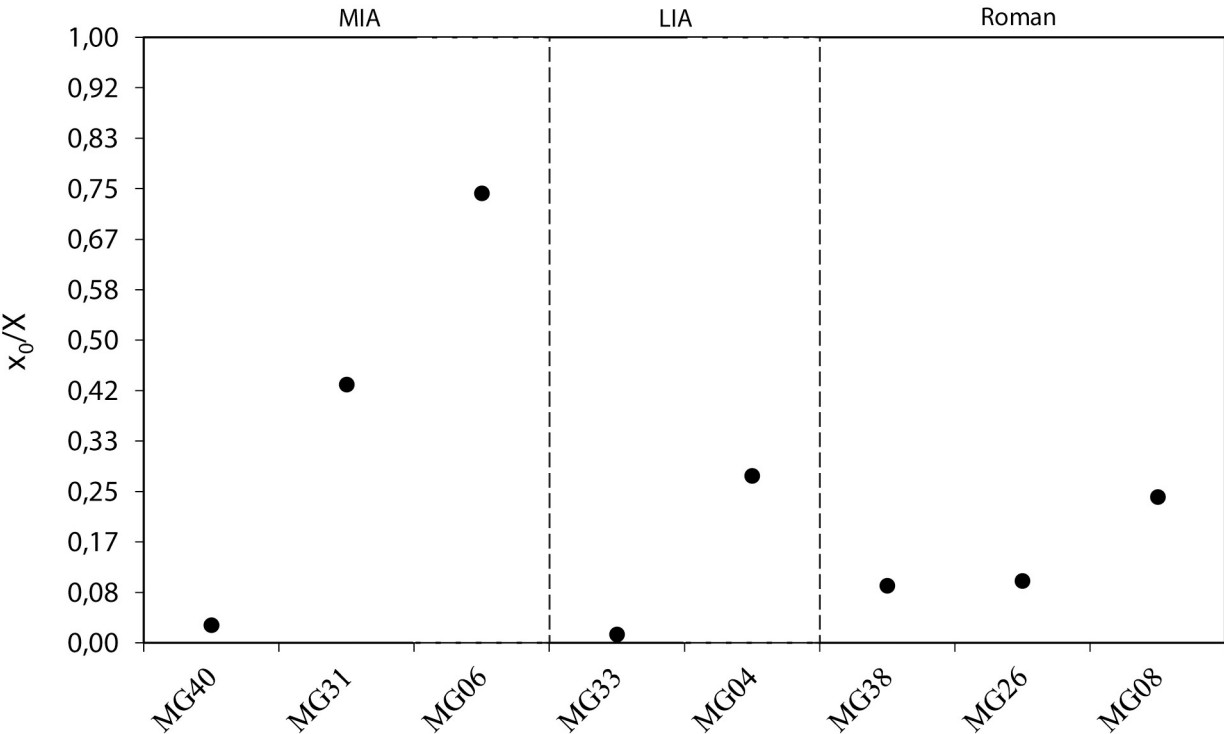

**Fig 8. Distribution of cattle birth based on the normalized data sets ($x_0/X$) from modelled δ18O M3 teeth at Houten-Castellum according to archaeological period.**

for this region before and is likely to be related to intensification of arable farming and not to an increase of dairy cows [12]. Sex ratios show a dominance of females and a slightly higher female to male ratio in the Middle Iron Age than in the Roman period. The higher proportion of calves killed in the Middle Iron Age compared to a slightly higher proportion of cows fits with the longer birth season and can tentatively be understood as a stronger emphasis on milk in the Middle Iron Age compared to the later periods. The lack of clear signs of specialised milk production can be explained by the fact that herds were kept in a mixed farming, subsistence system. The use of milk occurred at a small scale (household level) and was one of several products cattle provided. Spreading out births ensured that milk was available year-round. In non-specialised subsistence farming, sequential analysis of $\delta^{18}$O from carbonate in tooth enamel may be a favourable indicator for the targeted use of milk.

## Cattle diet

The variety in carbon isotopic composition for the ten cattle that were included in this study is low. Excluding non-local MG04, which has much lower values, as well as the single deviating value from MG13, the range of values is 1.5‰. Even when these lower values are included, the range is 2.5‰, which is much less than that reported for eight cattle teeth from a Neolithic site in the west of the Netherlands, where the range is 3.5‰ [50]. The intra-tooth amplitude of variation is also low: the eight animals consistent with local origin have amplitudes of 0.2 to 0.9‰, while only the two non-local cattle have higher amplitudes of 1.2 and 2.3‰. The amplitudes for the local animals are comparable with those found in third molars from Chillingham cattle, which are feral cattle living outside year-round and are fed hay in winter [67]. While a reduced amplitude of intra-tooth variation in cattle compared to sheep seems to be typical and may be related to the longer duration of mineralization in cattle [68], some cattle do show significantly higher amplitudes (e.g. our non-local animals, but also cattle from Neolithic and Chalcolithic sites in the Netherlands, Greece and Romania [50, 68–71]).

Most $\delta^{13}$C values fall within the expected range of -13.2 to -11.3‰. The two exceptions are MG04 and one value for MG13. The range translates to calculated plant values with a mean of -26.2‰ (excluding MG04 and the low value for MG13; based on +14.1‰ δ13C enrichment between diet and enamel bioapatite [38]). This corresponds clearly to a diet dominated by $C_3$ plants. It was not expected that $C_4$ plants would have contributed to cattle diet, as there are few indigenous $C_4$ plants in the Netherlands. The $C_4$ crop millet was grown in the Netherlands, but only in small quantities compared to other crops and is unlikely to have been used as fodder. The results confirm that $C_4$ plants are insignificant to cattle diet in our region during the Iron Age and Roman period.

There is also no evidence for a canopy effect, which occurs when animals graze in dense forest or are fed leaf fodder collected in dense forest. Again, this is not unexpected, since archaeobotanical research has shown that there was only limited forest in our region. Although non-local MG04 and the lowest value for MG13 have lower $\delta^{13}$C values than the other teeth (minima of -13.86‰ and -13.76‰, respectively), these values are nowhere near the lowest values reported by Balasse and colleagues for Neolithic Bercy, France (-14.7‰) [47].

The deviating low values of MG04 can be explained by the fact that this animal was raised in a different location (as strontium isotope analysis has shown [8]) and must have consumed a different diet and/or lived in a different environment. The strontium isotope ratio for MG13 is consistent with the local signal. The sample for strontium was taken in the same location of the tooth where the deviating $\delta^{13}$C value was found. However, the area with this particular strontium ratio is large and it is possible that the animal was living at another location with a similar geology but different vegetation at the time of formation of this part of the tooth.

In cattle grazing in the same pasture year-round, $\delta^{13}C$ sequences are expected to co-vary with $\delta^{18}O$ sequences, as vegetation will show seasonal variety in $\delta^{13}C$ values. A lack of correlation between $\delta^{13}C$ and $\delta^{18}O$ sequences suggests that cattle did not graze in the same location year-round; they were either moved between pastures and/or received fodder, as was suggested for cattle from Pool, Orkney and the Upper Rhine Valley [43, 44]. Feeding animals $^{13}C$-enriched fodder (enriched because collected in summer) during winter would lead to higher values during the winter month, disturbing normal seasonal variation in $\delta^{13}C$ sequences [41, 42].

For five of the teeth from Houten-Castellum (MG04, MG08, MG33, MG38 and MG40), the $\delta^{13}C$ sequences seem to co-vary with the $\delta^{18}O$ curves, although they are much less pronounced. One tooth from Houten-Castellum has a $\delta^{13}C$ sequence that hardly fluctuates (MG24, amplitude of variation of 0.21‰). Unfortunately, we have no modern data on free-ranging cattle in our region to compare with, so we do not know how much seasonal variation to expect. $\delta^{13}C$ curves for most of the cattle from Neolithic Schipluiden are also much less pronounced than the $\delta^{18}O$ curves [50], so perhaps this is typical for the Netherlands. The dampened $\delta^{13}C$ curves may indicate that these cattle received some kind of fodder during the period when the tooth was formed (second year of life), but at the moment we cannot exclude the possibility that moving between different pastures would have the same result. Grazing cattle in flood basins (including marsh vegetation) in summer may have led to a $^{13}C$-depleted diet, while grazing cattle on stubble and fallow fields in winter may have led to a $^{13}C$-enriched diet, which theoretically could cancel out naturally occurring differences in $\delta^{13}C$ values in vegetation.

Tooth MG06 has a $\delta^{13}C$ sequence that appears to show natural seasonal changes in vegetation by following to some extent the $\delta^{18}O$ curve, except for one sample location, where the $\delta^{13}C$ value is high while the $\delta^{18}O$ value is low. This could indicate that the animal was given $^{13}C$-enriched fodder (such as hay) in winter. In tooth MG26, the $\delta^{13}C$ sequence does not follow the $\delta^{18}O$ curve. Provision with fodder may have caused the lack of correlation. Indeed, one of the lowest $\delta^{18}O$ values coincide with one of the highest $\delta^{13}C$ values, possibly suggesting that this animal received $^{13}C$-enriched fodder in the winter.

Tooth MG31 also has a $\delta^{13}C$ sequence that does not follow the $\delta^{18}O$ curve. This individual has a non-local strontium signal at the beginning of M3 tooth formation. Unfortunately, we do not know when the animal moved to Houten-Castellum. If this occurred during the second year of life, then perhaps the change to a different environment with different vegetation can explain the lack of correlation between the $\delta^{13}C$ and $\delta^{18}O$ sequences. Finally, MG13, the tooth with one very low $\delta^{13}C$ value, does not have a complete $\delta^{18}O$ curve, but it is striking that the lowest $\delta^{13}C$ value is found in the location with the highest $\delta^{18}O$ value. As mentioned above, perhaps at the time when this part of the tooth was formed this animal lived in another part of the area with the 'local' strontium ratio and consumed vegetation depleted in $^{13}C$ compared to the vegetation consumed by most of the cattle in Houten-Castellum.

These results tentatively suggest that cattle in Houten-Castellum received fodder during their second year of life. However, we need comparative data from modern free-ranging cattle or other archaeological assemblages to confirm whether the lack of seasonal variation is typical for our region or whether it represents foddering. Lacking such data but considering the archaeological and archaeobotanical indications for the use of fodder, we can conclude that foddering was practised in the Iron Age and Roman Netherlands.

## Cattle drinking water

Sequential sampling of $\delta^{18}O$ along the tooth crown was expected to show a sinusoidal curve, reflecting seasonal variation in precipitation, with the highest values occurring in the summer

and the lowest in the winter [32, 69, 72–77]. The amplitude of the $\delta^{18}$O curves in the cattle teeth from Houten-Castellum is relatively low. While the lowest amplitudes are comparable to the lowest values found in Schipluiden [50], more of the teeth from Houten-Castellum have low amplitudes. The teeth with the highest amplitudes include two teeth that are from non-local cattle, so this may say nothing about local conditions.

Depressed amplitudes can have different causes. The first—movement to higher altitudes or other areas with relatively lower $\delta^{18}$O values in the summer [78–80]–is highly unlikely for the Netherlands. The second—a homogenised water source that is not controlled by seasonal rainwater fluctuations (such as an aquifer or large lake) [73, 74]–also seems highly unlikely for our region. The availability of drinking water for livestock would not have been a problem in the river area. Cattle could have drunk from rivers, standing water in the lowest parts of the flood basins and residual channels such as the one adjacent to the settlement of Houten-Castellum. Cattle hoof prints on the muddy banks of a residual channel in Geldermalsen-Hondsgemet suggest that such channels were used to water livestock [81]. Groundwater shows reduced $\delta^{18}$O seasonal changes because it averages $\delta^{18}$O values from precipitation over a longer time period [77]. Wells are common in Iron Age and Roman settlements in our region and perhaps we should consider the possibility that drinking water for stabled cattle was drawn from wells.

Third, differences between species in physiology, metabolism, diet and enamel mineralization rates may result in dampening of isotopic sequencing in cattle compared to other species [16, 68, 70, 73, 74, 78, 82, 83]. Low amplitudes have also been found in archaeological and modern cattle teeth from Great Britain [17, 43, 84, 85], while higher amplitudes occur in cattle teeth from archaeological sites in Greece and Romania as well as in northwestern Europe (Great Britain, France and Germany [44, 45, 47, 67, 84]). This suggests that low amplitudes are not typical for cattle in general.

## Conclusion

Although the sample size of ten teeth for three time periods is small, the stable isotope analysis has yielded interesting results. Sequential sampling of oxygen isotopes suggests a prolonged birth season in the Middle Iron Age. There are hints that the birth season was shorter in the Late Iron Age and Roman period but this will need to be confirmed by analysing additional samples. By combining the isotopic evidence with data from archaeological, archaeozoological and residue analysis, we can conclude that milk was used, and that milk was more important in the Middle Iron Age than in the later periods. Milk was, however, only one of several products that cattle were exploited for, which is why clear archaeozoological signals for specialised milk production are rare and organic residue analysis has not found dairy in all sites studied. Dairy exploitation was not the main focus of cattle husbandry, but rather occurred at household level within a self-sufficient farming system. If there was indeed a decreased prominence of dairy exploitation in the Late Iron Age and Roman paper, this can perhaps be related to an increased emphasis on other products, such as meat and traction, stimulated by the demand for food by the Roman urban and military markets. Sequential sampling of carbon isotopes suggests that the provision of fodder to subadult cattle was practised in the Iron Age and Roman period. This confirms results from archaeological and archaeobotanical research. The data set in this study is small but the current dearth of parallel evidence from the Netherlands means that, although the results are tentative, they have the potential to stimulate further research in this area.

## Supporting information

**S1 Table. Context information on the specimens used in this study.**
(PDF)

**S2 Table. Intra-tooth oxygen and carbon isotope ratios of enamel from cattle teeth from Houten-Castellum.** Wear stages after Grant (1982) [23].
(PDF)

## Acknowledgments

We would like to thank Hilary Sloane (NERC Isotope Geosciences Laboratory, British Geological Survey) for running the samples. We would also like to thank Mirella de Jong (Provinciaal Depot voor Bodemvondsten Utrecht) for permission to sample teeth and Martijn van Haasteren, Edda Wijnans and Jan van Renswoude (VUhbs) for their help in finding the teeth.

## Author Contributions

**Conceptualization:** Maaike Groot, Umberto Albarella, Jane Evans.

**Data curation:** Maaike Groot.

**Formal analysis:** Maaike Groot, Jana Eger, Jane Evans.

**Funding acquisition:** Maaike Groot, Umberto Albarella.

**Investigation:** Maaike Groot.

**Methodology:** Maaike Groot, Jana Eger, Jane Evans.

**Project administration:** Maaike Groot.

**Supervision:** Umberto Albarella, Jane Evans.

**Visualization:** Maaike Groot, Jana Eger.

**Writing – original draft:** Maaike Groot.

**Writing – review & editing:** Maaike Groot, Umberto Albarella, Jana Eger, Jane Evans.

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
