## [Decision Letter · Decision Letter 0]

16 Aug 2021

PONE-D-21-23375

Cattle management in an Iron Age/Roman settlement in the Netherlands: archaeozoological and stable isotope analysis

PLOS ONE

Dear Dr. Groot,

Thank you for submitting your manuscript to PLOS ONE. After careful consideration, we feel that it has merit but does not fully meet PLOS ONE’s publication criteria as it currently stands. Therefore, we invite you to submit a revised version of the manuscript that addresses the points raised during the review process.

All comments need to be addressed in detail.

We look forward to receiving your revised manuscript.

Kind regards,

Peter F. Biehl, PhD

Academic Editor

PLOS ONE

Journal Requirements:

2. In your manuscript, please provide additional information regarding the specimens used in your study. Ensure that you have reported specimen numbers and complete repository information, including museum name and geographic location.

For more information on PLOS ONE's requirements for paleontology and archaeology research, see https://journals.plos.org/plosone/s/submission-guidelines#loc-paleontology-and-archaeology-research.

Additional Editor Comments:

Please address all comments.

Reviewers' comments:

Reviewer's Responses to Questions

**Comments to the Author**

1. Is the manuscript technically sound, and do the data support the conclusions?

Reviewer #1: Yes

2. Has the statistical analysis been performed appropriately and rigorously? 

Reviewer #1: Yes

3. Have the authors made all data underlying the findings in their manuscript fully available?

Reviewer #1: Yes

4. Is the manuscript presented in an intelligible fashion and written in standard English?

Reviewer #1: Yes

5. Review Comments to the Author

Reviewer #1: I recommend this excellent research for publication, with a few suggestions that are at the discretion of the authors. The piece stands well as it is.

My comments:

Clear, informative, and novel

Situates this site into the developing understanding of animal rearing, seasonality, and landscape in the region

Nicely describes relevant background during the introduction but stays focused

Specific enough for specialists but still described clearly- could avoid some words such as hypsodont, etc. but overall very clear., especially the isotope descriptions. The article is primarily interesting for a specialist audience but clear enough that non-specialists can understand the implications of the research

Cites ample and relevant studies, identifies gaps

Clearly describes methods, follows standard procedures

Page 8 Line 97: authors meaning of the previous study or this paper (or perhaps they are the same)?

Page 10 Line 250: adding the word “either” before cheese and a comma after herb could help clarify this sentence

Page 10 Line 254: suggestion of an additional sentence summarizing the organic residue evidence for dairy—present but not ubiquitous? Not well enough studied to establish a pattern? This could be highlighted in the discussion-residue and archaeozoological remains indicate some milk exploitation, but it was not the focus? Re-establishing these findings within the new isotope findings might be impactful

If there is space, I would like to see it come back to self-sufficiency and household-level exploitation of milk, or some ideas about why there was a shift towards decreased birthing season/prominence of milk exploitation at the end of the paper, or why this study matters more broadly. Take the conclusion one step bolder and broader

6. PLOS authors have the option to publish the peer review history of their article (what does this mean?). If published, this will include your full peer review and any attached files.

Reviewer #1: No

---

## [Author Response · Author response to Decision Letter 0]

25 Aug 2021

Academic editor’s comments:

-I have checked the manucript and file names and hope they now meet the requirements.

2. In your manuscript, please provide additional information regarding the specimens used in your study. Ensure that you have reported specimen numbers and complete repository information, including museum name and geographic location.

-I have added a table in the supplementary information with additional information on the specimens used.

-Figure 1 has been replaced. 

-I have checked my reference list and I believe it is complete. I do not think I have cited any retracted articles.

Reviewer’s comments:

could avoid some words such as hypsodont

-I have added a brief explanation of ‘hypsodont’ the first time it is mentioned in the text. 

Page 8 Line 97: authors meaning of the previous study or this paper (or perhaps they are the same)?

-I have changed the text to “authors of the Schipluiden study” to clarify this.

Page 10 Line 250: adding the word “either” before cheese and a comma after herb could help clarify this sentence

-done

Page 10 Line 254: suggestion of an additional sentence summarizing the organic residue evidence for dairy—present but not ubiquitous? Not well enough studied to establish a pattern? This could be highlighted in the discussion-residue and archaeozoological remains indicate some milk exploitation, but it was not the focus? Re-establishing these findings within the new isotope findings might be impactful.

-I have added two sentences to summarize the organic residue evidence, as suggested. I have also changed the text in the discussion (“Birth season”) and conclusion somewhat to emphasise the point about organic residue. The lack of a focus on milk as shown by the archaeozoological remains already seems clear enough to me as the text is now. 

If there is space, I would like to see it come back to self-sufficiency and household-level exploitation of milk, or some ideas about why there was a shift towards decreased birthing season/prominence of milk exploitation at the end of the paper, or why this study matters more broadly. Take the conclusion one step bolder and broader.

-I have added a few sentences in the discussion on this issue.

---

## [Decision Letter · Decision Letter 1]

22 Sep 2021

Cattle management in an Iron Age/Roman settlement in the Netherlands: archaeozoological and stable isotope analysis

PONE-D-21-23375R1

Dear Dr. Groot,

We’re pleased to inform you that your manuscript has been judged scientifically suitable for publication and will be formally accepted for publication once it meets all outstanding technical requirements.

Kind regards,

Peter F. Biehl, PhD

Academic Editor

PLOS ONE

Additional Editor Comments (optional):

Reviewers' comments:

Reviewer's Responses to Questions

**Comments to the Author**

1. If the authors have adequately addressed your comments raised in a previous round of review and you feel that this manuscript is now acceptable for publication, you may indicate that here to bypass the “Comments to the Author” section, enter your conflict of interest statement in the “Confidential to Editor” section, and submit your "Accept" recommendation.

Reviewer #1: All comments have been addressed

2. Is the manuscript technically sound, and do the data support the conclusions?

Reviewer #1: Yes

3. Has the statistical analysis been performed appropriately and rigorously? 

Reviewer #1: Yes

4. Have the authors made all data underlying the findings in their manuscript fully available?

Reviewer #1: Yes

5. Is the manuscript presented in an intelligible fashion and written in standard English?

Reviewer #1: Yes

6. Review Comments to the Author

Reviewer #1: All comments were addressed. This article is a substantial contribution to the field and follows rigorous scientific standards, explained in clear terms and connected to broader impacts.

7. PLOS authors have the option to publish the peer review history of their article (what does this mean?). If published, this will include your full peer review and any attached files.

Reviewer #1: No

---

## [Editor Report · Acceptance letter]

24 Sep 2021

PONE-D-21-23375R1 

Cattle management in an Iron Age/Roman settlement in the Netherlands: archaeozoological and stable isotope analysis 

Dear Dr. Groot:

I'm pleased to inform you that your manuscript has been deemed suitable for publication in PLOS ONE. Congratulations! Your manuscript is now with our production department. 

Kind regards, 

on behalf of

Dr. Peter F. Biehl 

Academic Editor

PLOS ONE